# Microtubule assembly by tau impairs endocytosis and neurotransmission via dynamin sequestration in Alzheimer's disease synapse model

**Tetsuya Hori[1,2], Kohgaku Eguchi[1,3], Han-Ying Wang[1], Tomohiro Miyasaka[2], Laurent Guillaud[1,4], Zacharie Taoufiq[1], Satyajit Mahapatra[1], Hiroshi Yamada[5], Kohji Takei[5], Tomoyuki Takahashi[1]\***

[1]Cellular and Molecular Synaptic Function Unit, Okinawa Institute of Science and Technology - Graduate University, Okinawa, Japan; [2]Faculty of Life and Medical Sciences, Doshisha University, Kyoto, Japan; [3]Institute of Science and Technology Austria (IST Austria), Klosterneuburg, Austria; [4]Molecular Neuroscience Unit, Okinawa Institute of Science and Technology Graduate University, Okinawa, Japan; [5]Department of Neuroscience, Graduate school of Medicine, Dentistry and Pharmaceutical Sciences, Okayama University, Okayama, Japan

**Abstract** Elevation of soluble wild-type (WT) tau occurs in synaptic compartments in Alzheimer's disease. We addressed whether tau elevation affects synaptic transmission at the calyx of Held in slices from mice brainstem. Whole-cell loading of WT human tau (h-tau) in presynaptic terminals at 10–20 µM caused microtubule (MT) assembly and activity-dependent rundown of excitatory neurotransmission. Capacitance measurements revealed that the primary target of WT h-tau is vesicle endocytosis. Blocking MT assembly using nocodazole prevented tau-induced impairments of endocytosis and neurotransmission. Immunofluorescence imaging analyses revealed that MT assembly by WT h-tau loading was associated with an increased MT-bound fraction of the endocytic protein dynamin. A synthetic dodecapeptide corresponding to dynamin 1-pleckstrin-homology domain inhibited MT-dynamin interaction and rescued tau-induced impairments of endocytosis and neurotransmission. We conclude that elevation of presynaptic WT tau induces de novo assembly of MTs, thereby sequestering free dynamins. As a result, endocytosis and subsequent vesicle replenishment are impaired, causing activity-dependent rundown of neurotransmission.

## Editor's evaluation

This study provides an interesting new insight into the synaptic disease mechanisms of tauopathies. The paper is based on a technically very rigorous dataset indicating that increased levels of soluble Tau impair pre-synaptic endocytosis and, consequently, neurotransmission by sequestering Dynamin-1 on microtubules. The findings are of major relevance for basic neuronal cell biology and translational neuroscience alike.

## Introduction

The microtubule (MT)-binding protein tau assembles and stabilizes MTs (*Brunden et al., 2009*; *Lee et al., 2011*) mainly in axonal compartments (*Binder et al., 1985*; *Kubo et al., 2019*). Phosphorylation of tau proteins reduces their binding affinity (*Biernat et al., 1993*; *Drechsel et al., 1992*), thereby

**\*For correspondence:**
ttakahas@oist.jp

**Competing interest:** The authors declare that no competing interests exist.

shifting the equilibrium from MT-bound form to soluble free form (*Ballatore et al., 2007*). Soluble tau proteins also exist in dynamic equilibrium between phosphorylated and dephosphorylated forms (*Spillantini and Goedert, 2013*) as well as between soluble and aggregated forms. When the cytosolic tau concentration is elevated, monomeric tau undergoes oligomerization and eventually precipitates into neurofibrillary tangles (NFT) (*Greenberg and Davies, 1990*; *Grundke-Iqbal et al., 1986*; *Yoshida and Ihara, 1993*), which is a hallmark of tauopathies, including Alzheimer's disease (AD), frontotemporal dementia with Parkinsonism-17 (FTDP-17), and progressive supranuclear palsy (*Lee et al., 2011*; *Ballatore et al., 2007*; *Spillantini and Goedert, 2013*). Although the NFT density can correlate with the degree of AD progression (*Lee et al., 2011*; *Ballatore et al., 2007*; *Morris et al., 2011*), soluble tau protein levels are more closely linked to disease progression and cognitive decline (*Götz et al., 2008*; *Koss et al., 2016*).

Genetic ablation of tau shows little abnormal phenotype (*Harada et al., 1994*; *Vossel et al., 2010*; *Yuan et al., 2008*), presumably due to compensation by other MT-associated proteins (*Harada et al., 1994*). Instead, tau ablation can prevent amyloid β-induced impairments of mitochondrial transport (*Vossel et al., 2010*) or memory defects (*Ittner et al., 2010*; *Roberson et al., 2007*). Thus, loss of tau function due to its dissociation from MTs is unlikely to be an important cause of neuronal dysfunction in AD (*Spillantini and Goedert, 2013*; *Morris et al., 2011*).

In postmortem brains of both AD patients and intact humans, tau is present in synaptosomes (*Fein et al., 2008*; *Tai et al., 2012*). In a transgenic mice AD model, soluble tau is accumulated in the hippocampal nerve terminal zone (*de Calignon et al., 2012*; *Liu et al., 2012*). Both in in vivo and in culture models of tauopathy, tau is released from axon terminals upon KCl stimulation in a Ca$^{2+}$-dependent manner, like neurotransmitters (*Pooler et al., 2013*; *Yamada et al., 2014*). Tau oligomers produced by released tau triggers endogenous tau seeding in neighboring neurons, thereby causing trans-synaptic propagations (*de Calignon et al., 2012*; *Guo and Lee, 2011*).

FTDP tauopathy model mice that overexpressed with mutant tau are widely used to examine tau toxicities on synaptic plasticity (*Polydoro et al., 2014*; *Sydow et al., 2011*; *Yoshiyama et al., 2007*), memory formation (*Sydow et al., 2011*; *Santacruz et al., 2005*), as well as on synaptic vesicle (SV) transport (*McInnes et al., 2018*; *Zhou et al., 2017*). In contrast to FTDP, which is a rare familial disease associated with tau mutation, AD is a widespread sporadic disease unassociated with tau mutation, but the expression level of WT tau being crucial. As AD models, the effects of WT tau overexpression have been examined in culture cells (*Ebneth et al., 1998*; *Shahpasand et al., 2012*; *Stamer et al., 2002*; *Thies and Mandelkow, 2007*) or in *Drosophila* (*Mudher et al., 2004*), where impaired axonal transports associated with increased MT density were found. These observations suggest that WT tau can be detrimental when its levels are elevated (*Stamer et al., 2002*; *Thies and Mandelkow, 2007*). However, unlike FTDP tau mutant, it is unknown whether elevated soluble WT tau can affect mammalian central synaptic transmission.

We addressed this question using the giant nerve terminal calyx of Held visualized in slices from mice brainstem, where axonal MTs extended into the depth of terminals (*Piriya Ananda Babu et al., 2020*). In this presynaptic terminal, we loaded recombinant WT h-tau from a whole-cell patch pipette at fixed concentrations to model the elevation of WT tau associated with AD and found that WT h-tau newly assembled MTs and strongly impaired synaptic transmission. Capacitance measurements indicated that the primary target of WT h-tau is vesicle endocytosis. Immunocytochemical image analysis after cell permeabilization revealed an increase in the MT-bound fraction of the endocytic GTPase dynamin in WT h-tau-loaded terminals. Since the endocytic key protein dynamin is an MT-binding protein (*Shpetner and Vallee, 1989*), dynamin is likely sequestered by newly assembled MTs. Out of screening, we found that a synthetic dodecapeptide corresponding to amino acids 560–571 of dynamin 1 inhibited MT-dynamin interaction. When we co-loaded this peptide 'PHDP5' with WT h-tau, its toxicities on vesicle endocytosis as well as on synaptic transmission were rescued. Thus, we propose a novel synaptic dysfunction mechanism underlying AD, in which WT tau-induced over-assembly of MTs depletes dynamins, thereby impairing vesicle endocytosis and synaptic transmission.

## Results

### Intra-terminal loading of WT h-tau impairs excitatory synaptic transmission

To address whether elevation of soluble h-tau in presynaptic terminals can affect synaptic transmission, we purified WT recombinant h-tau (0N4R) and its deletion mutant (del-MTBD) lacking the MT-binding site ($^{244}$Gln–$^{367}$Gly) (*Figure 1—figure supplement 1A*), obtained using an *Escherichia coli* expression system (*Xie et al., 2014*). These recombinant h-tau proteins are highly soluble at room temperature (RT) without any sign of granulation (*Maeda et al., 2007*). In simultaneous pre- and postsynaptic recording at the calyx of Held in mouse brainstem slices, we recorded EPSCs evoked at 1 Hz by presynaptic action potentials (*Figure 1*). After confirming stable EPSC amplitude for 10 min, we injected a large volume of internal solution containing WT h-tau (20 µM) from an installed fine tube to a presynaptic whole-cell pipette to replace most of pipette solution and allow h-tau to diffuse into a presynaptic terminal (illustration in *Figure 1A*; *Hori et al., 1999*; *Takahashi et al., 2012*). After loading h-tau (20 µM), the amplitude of glutamatergic EPSCs gradually declined and reached 23% ± 9% in 30 min (*Figure 1A*, p < 0.01, paired t-test, n = 6 synapses in 6 slices). WT h-tau loaded at a lower concentration (10 µM) caused a slower EPSC rundown to 65% ± 5% in 30 min (p < 0.01, n = 5 synapses in 5 slices). Del-MTBD (20 µM), lacking tubulin polymerization capability (*Figure 1—figure supplement 1B*), likewise loaded had no effect on EPSC amplitude (*Figure 1A*). Since h-tau concentrations in presynaptic terminals are equilibrated with those in a presynaptic whole-cell pipette with a much greater volume than terminals (*Pusch and Neher, 1988*), these results suggest that WT h-tau >10 µM can significantly impair excitatory synaptic transmission.

The inhibitory effect of WT h-tau on EPSCs was apparently frequency-dependent. When evoked at 0.1 Hz, WT h-tau (20 µM) caused only a minor reduction of EPSC amplitude (to 85% ± 12%, 30 min after loading, p = 0.21, n = 5; *Figure 1B*). Since taxol shares a common binding site of MTs with tau (*Kar et al., 2003*) and assembles tubulins into MTs (*Figure 1—figure supplement 1B*), we tested the effect of taxol (1 µM) on EPSCs (*Figure 1C*). Like h-tau, taxol caused a significant rundown of EPSCs evoked at 1 Hz (to 41 ± 12 at 30 min, n = 5, p < 0.05), but not those evoked at 0.1 Hz (104% ± 3.0%, n = 4, p = 0.60). These results together suggest that MTs newly assembled in presynaptic terminals by WT h-tau or taxol cause activity-dependent rundown of excitatory synaptic transmission.

### WT h-tau primarily inhibits SV endocytosis and secondarily exocytosis

To determine the primary target of h-tau causing synaptic dysfunction, we performed membrane capacitance measurements at the calyx of Held (*Eguchi et al., 2017*; *Sun and Wu, 2001*; *Wang et al., 2020*; *Yamashita et al., 2005*). Since stray capacitance of perfusion pipettes prevents capacitance measurements, we backfilled h-tau into a conventional patch pipette after preloading normal internal solution only at its tip to secure GΩ seal formation. This caused substantial and variable delays of the intra-terminal diffusion, so no clear effect could be seen more than 10 min after whole-cell patch membrane was ruptured. Twenty minutes after whole-cell patch loading of WT h-tau (20 µM), endocytic capacitance showed a significant slowing (*Figure 2*), whereas exocytic capacitance magnitude (ΔC$_m$) or charge of Ca$^{2+}$ currents (Q$_{Ca}$) induced by a depolarizing pulse was not different from controls without h-tau loading. Thirty minutes after loading h-tau, the endocytic capacitance change became further slowed (p < 0.01), and exocytic ΔC$_m$ eventually showed a significant reduction (p < 0.05, n = 5) without a change in Q$_{Ca}$. These results suggest that the primary target of h-tau toxicity is SV endocytosis. Endocytic block inhibits recycling replenishment of SVs via recycling, thereby reducing the exocytic release of neurotransmitter as a secondary effect.

### Inhibition of SV endocytosis and synaptic transmission by WT h-tau requires de novo MT assembly

Since new MT assembly might take place after h-tau loading (*Figure 1*, *Figure 1—figure supplement 1*), we tested whether the tubulin polymerization blocker nocodazole might reverse the toxic effects of h-tau on SV endocytosis and synaptic transmission. In tubulin polymerization assays, nocodazole inhibited h-tau-dependent MT assembly in a concentration-dependent manner, with a maximal inhibition reached at 20 µM (*Figure 3A*). In presynaptic capacitance measurements, nocodazole (20 µM) co-loaded with h-tau (20 µM) fully prevented the h-tau toxicities on endocytosis (*Figure 3B*) and

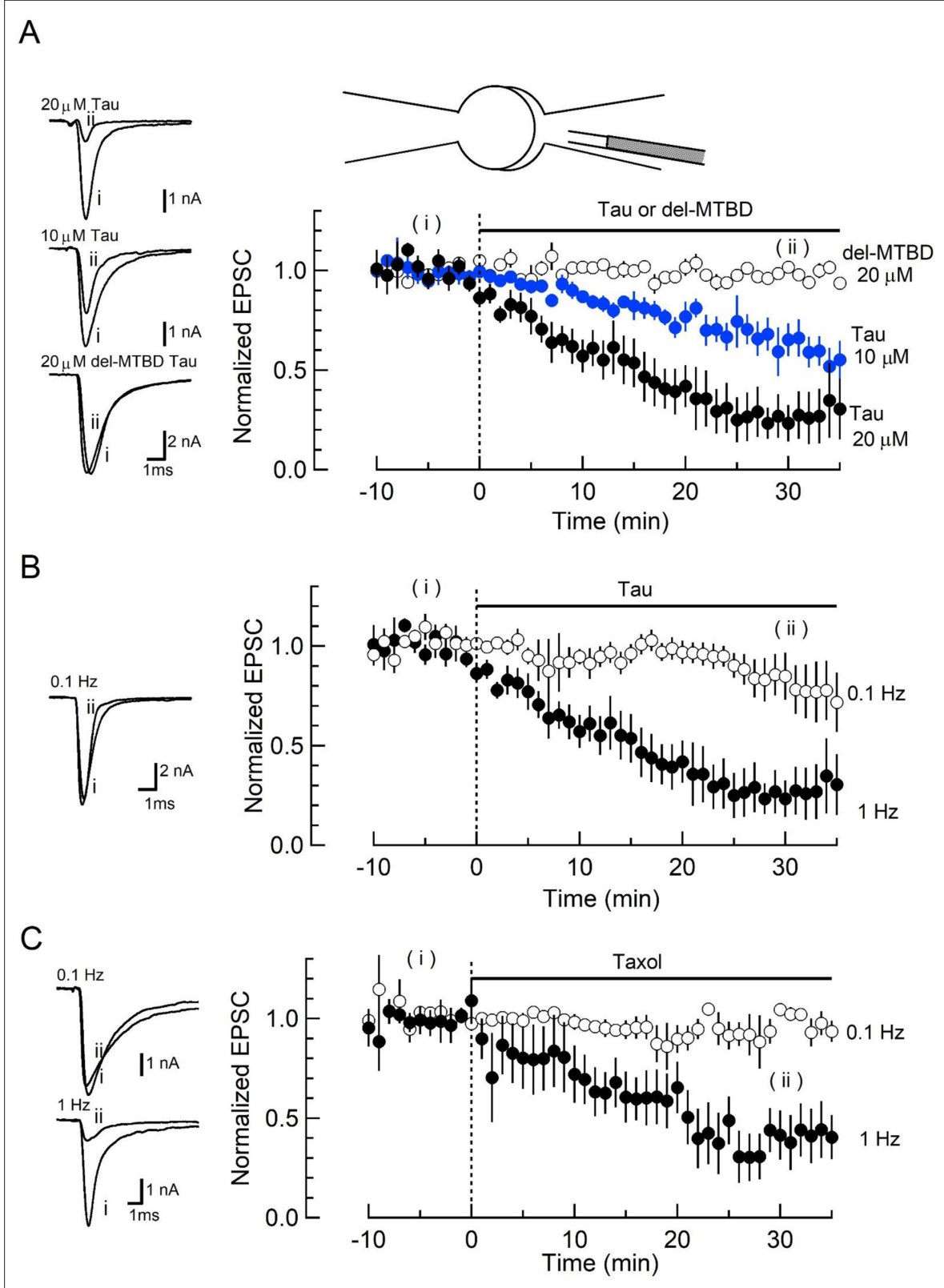

**Figure 1.** Wild-type (WT) human tau (h-tau) loaded in presynaptic terminals inhibited excitatory synaptic transmission. (**A**) In simultaneous pre- and postsynaptic whole-cell recordings, intra-terminal infusion of WT h-tau at 10 μM (blue filled circles) or 20 μM (black filled circles), from a tube in a presynaptic patch pipette (top illustration), caused a concentration-dependent rundown of EPSCs evoked by presynaptic action potentials at 1 Hz. In the time plots, EPSC amplitudes averaged from 60 events are sampled for data points and normalized to the mean amplitude of baseline EPSCs before

*Figure 1 continued on next page*

*Figure 1 continued*

h-tau infusion. Sample records of EPSCs 5 min before (i) and 30 min after (ii) tau infusion are superimposed and shown on the left panels. The EPSC amplitude remaining 30 min after infusion was 23% ± 9% and 65% ± 5%, respectively, for 10 and 20 µM h-tau (means and SEMs, 6 synapses from 6 slices, p < 0.01 in paired t-test between before and after h-tau infusion). Infusion of microtubule (MT)-binding site-deleted h-tau mutant (del-MTBD, 20 µM, *Figure 1—figure supplement 1A*) had no effect on the EPSC amplitude (open circles, sample EPSC traces shown on the left bottom panel). (**B**) The amplitude of EPSCs evoked at 0.1 Hz remained unchanged after h-tau infusion (85% ± 12%, 5 synapses from 5 slices, p = 0.22 in paired t-test). Sample records of EPSCs before (i) and 30 min after (ii) h-tau infusion at 0.1 Hz are superimposed on the left panel. (**C**) Taxol (1 µM) caused activity-dependent rundown of EPSC amplitude to 41.4% ± 12% at 1 Hz (p < 0.01, 5 synapses from 5 slices), but remained unchanged when stimulated at 0.1 Hz (105% ± 3.0%, open circles, 4 synapses from 4 slices). Sample records of EPSCS at 0.1 and 1 Hz are superimposed on the left panels.

The online version of this article includes the following source data and figure supplement(s) for figure 1:

**Source data 1.** Wild-type (WT) h-tau loaded in presynaptic terminals inhibited excitatory synaptic transmission.

**Figure supplement 1.** Tubulin polymerization assay for purified 0N4R wild-type (WT) human tau (h-tau) and microtubule (MT)-binding region-deleted mutant.

**Figure supplement 1—source data 1.** Raw SDS-PAGE gel data from *Figure 1—figure supplement 1A*.

**Figure supplement 1—source data 2.** Data from *Figure 1—figure supplement 1B*.

synaptic transmission (*Figure 3C*). Nocodazole alone (20 µM) had no effect on exo-endocytosis (*Figure 3B*) or EPSC amplitude (*Figure 3C*). It is highly likely that WT h-tau loaded in calyceal terminals newly assembled MTs, thereby impairing SV endocytosis and synaptic transmission.

## WT h-tau assembles MTs and sequesters dynamins in calyceal terminals

The monomeric GTPases dynamins 1 and 3 play critical roles in the endocytic fission of SVs (*Hinshaw and Schmid, 1995*; *Raimondi et al., 2011*; *Takei et al., 1995*). Since dynamin is originally discovered as an MT-binding protein (*Shpetner and Vallee, 1989*), we hypothesized that newly assembled MTs might trap free dynamins in cytosol. If this is the case, MT-bound form of dynamin would be increased. To test this hypothesis, we performed immunofluorescence microscopy and image analysis to quantify MTs and dynamin. After whole-cell infusion of h-tau into calyceal terminals, slices were chemically fixed and permeabilized to allow cytosolic-free molecules such as tubulin monomers to be washed out of the terminal, thereby enhancing the signals from large structures such as MTs or MT-bound molecules. Fluorescent h-tau antibody identified calyceal terminals loaded with WT h-tau (20 µM, *Figure 4A*). Double staining with mouse β3-tubulin antibody revealed a 2.1-fold increase in MT signals in h-tau-loaded terminals, compared with those without h-tau loading (p = 0.01, n = 5, two-tailed unpaired t-test with Welch's correction, *Figure 4B*). Triple labeling with dynamin antibodies further revealed a 2.6-fold increase in dynamin signal (p = 0.01, n = 5, two-tailed t-test with Welch's correction, *Figure 4B*). In super-resolution imaging, dynamins are shown in clusters along MTs in tau-loaded calyceal terminal (*Figure 4—figure supplement 1*). These results suggest that soluble WT h-tau can assemble MTs in presynaptic terminals, thereby sequestering cytosolic dynamins that are indispensable for SV endocytosis.

Besides dynamins, MTs can bind to various other proteins. Among them, formin mDia can bind to MTs (*Bartolini and Gundersen, 2010*) and involved in the endocytic scaffold functions together with F-actin, intersectin, and endophilin. Although acute depolymerization of F-actin (*Piriya Ananda Babu et al., 2020*; *Eguchi et al., 2017*) or genetic ablation of intersectin (*Sakaba et al., 2013*) has no effect on SV endocytosis at the calyx of Held, the formin mDia inhibitor SMFH2 reportedly inhibits endocytosis at the calyx terminals in pre-hearing rats (postnatal days [P] 8–12) (*Soykan et al., 2017*). We re-examined whether the drug might inhibit SV endocytosis at calyceal terminals in slices from post-hearing mice (P13–14). SMFH2 slightly prolonged SV endocytosis, but this effect was statistically insignificant (*Figure 4—figure supplement 2A*). Thus, formin unlikely makes substantial contribution to the marked endocytic slowing observed after intra-terminal tau loading (*Figure 2*).

It may also be argued that binding of endophilin to MTs (*Schuske et al., 2003*) might cause EPSC rundown since endophilin is involved in clathrin uncoating (*Watanabe et al., 2018*), which is required for SV refilling with glutamate. If SV refilling during recycling is impaired, miniature EPSCs are decreased in amplitude and frequency (*Takami et al., 2017*). However, neither amplitude nor frequency was affected by intra-terminal loading of tau (20 µM) (*Figure 4—figure supplement 2B*). Thus, endophilin-MT binding unlikely underlies EPSC rundown by intra-terminal tau infusion (*Figure 1*).

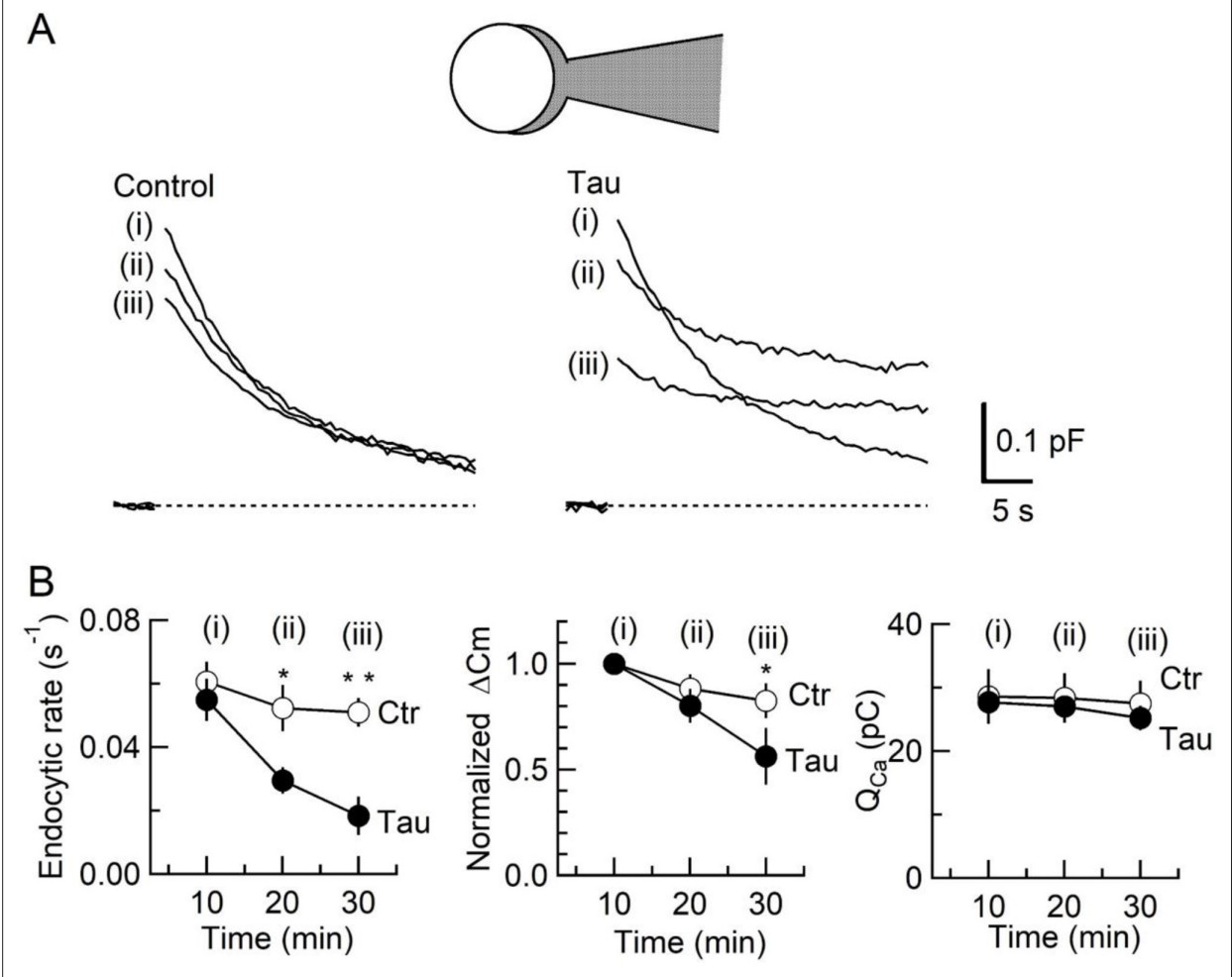

**Figure 2.** Inhibition of synaptic vesicle (SV) endocytosis is the primary effect of wild-type (WT) human tau (h-tau) loading. (**A**) Exo-endocytic membrane capacitance changes in presynaptic terminals without (Control) or after direct loading of WT h-tau (20 μM; Tau). WT h-tau was directly loaded by diffusion into a terminal from a whole-cell patch pipette (illustration). Capacitance traces were sampled from (**i**) 10, (**ii**) 20, and (**iii**) 30 min after patch membrane rupture (superimposed). *Left panel*, non-loading control. *Right panel*, WT h-tau-loaded terminal. Capacitance changes were evoked every 2 min by $Ca^{2+}$ currents induced by a 20 ms depolarizing pulse (not shown). (**B**) Time plots of endocytic rate (left panel), exocytic magnitude (middle panel), and presynaptic $Ca^{2+}$ current charge (right panel). Data points represent averaged values from five events from 4 min before and 4 min after the time points. In calyceal terminals, 20 min after patch membrane rupture with a pipette containing WT h-tau (filled circles; Tau), endocytic rate was significantly prolonged (*$p < 0.05$ compared to controls, open circles, repeated-measures two-way ANOVA with post hoc Scheffe test, n = 5 from 5 slices), whereas exocytic magnitude remained similar to controls ($p = 0.45$). Thirty minutes after rupture, endocytic rate was further prolonged (**$p < 0.01$) and exocytic magnitude became significantly less than controls (*$p < 0.05$). $Ca^{2+}$ current charge ($Q_{Ca}$) remained unchanged throughout recording.

The online version of this article includes the following source data for figure 2:

**Source data 1.** Inhibition of synaptic vesicle (SV) endocytosis is the primary effect of wild-type (WT) human tau (h-tau) loading.

## An MT-dynamin-binding inhibitor peptide attenuates h-tau toxicities on SV endocytosis and synaptic transmission

To prevent toxic effects of h-tau on endocytosis and transmission, we searched for a dominant-negative (DN) peptide blocking MT-dynamin binding. Since the MT-binding domain of dynamins is unknown, we synthesized 11 peptides from the pleckstrin-homology (PH) domain and 11 peptides from the proline-rich domain of dynamin 1 (*Figure 4—figure supplement 1A*) and submitted them to the MT-dynamin 1-binding assay. Out of 22 peptides, one peptide corresponding to the amino acid sequence 560–571 of PH domain, which we named 'PHDP5', significantly inhibited the MT-dynamin 1 interaction (*Figure 5*, *Figure 5—figure supplement 1B and C*). By SYPRO orange staining, dynamin 1 is found as an ~100 kDa band, 1.7% ± 0.4% in precipitates (ppts). In the presence of MT, dynamin 1

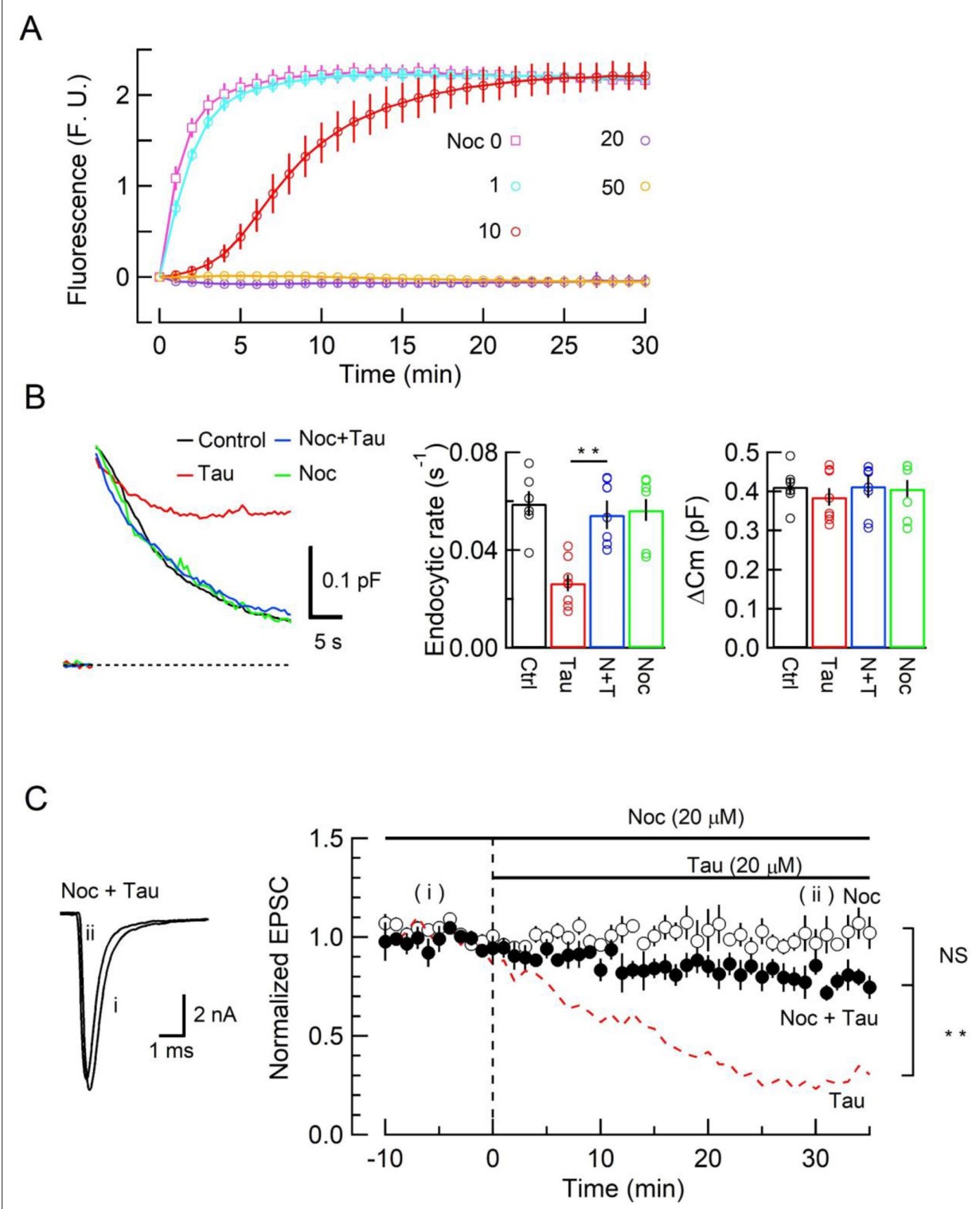

**Figure 3.** The microtubule (MT) assembly blocker nocodazole prevented tau-induced block of synaptic vesicle (SV) endocytosis and EPSC rundown. (**A**) Concentration-dependent inhibitory effects of nocodazole on MT assembly in tubulin polymerization assay. MT assembly by 0N4R human tau (h-tau) (20 μM) in the absence (pink symbols and a fitting line) or presence of nocodazole at 1 μM (blue), 10 μM (red), 20 μM (purple), and 50 μM (orange). Data points and error bars in all graphs represent means and SEMs (n = 3). (**B**) Nocodazole prevented h-tau-induced block of SV endocytosis. Presynaptic membrane capacitance changes (superimposed traces) 25 min after loading h-tau alone (20 μM, red trace), h-tau and nocodazole (20 μM, blue), nocodazole alone (20 μM, green), and controls with no loading (black). Bar graphs indicate endocytic rates in non-loading controls (Ctr, black, 6

*Figure 3 continued on next page*

*Figure 3 continued*

terminals from 6 slices), h-tau-loaded terminals (Tau, red, 8 terminals from 8 slices), co-loading of nocodazole with h-tau (N + T, blue, 7 terminals from 7 slices), and nocodazole alone (Noc, green, 8 terminals from 8 slices). Nocodazole co-loading fully prevented endocytic block by h-tau (**p < 0.01, between Tau and N-T) to control level (one-way ANOVA with Scheffe post hoc test). (**C**) Nocodazole prevented EPSC rundown caused by wild-type (WT) h-tau. Nocodazole (20 µM) co-loaded with WT h-tau (20 µM) prevented EPSC rundown (filled circles, 4 synapses from 4 slices, **p < 0.01, unpaired t-test). Data of WT h-tau effect on EPSCs (**Figure 1A**) is shown as a red dashed line for comparison. Nocodazole alone (20 µM) had no effect on EPSC amplitude throughout (open circles, 4 synapses from 4 slices).

The online version of this article includes the following source data for figure 3:

**Source data 1.** The microtubule (MT) assembly blocker nocodazole prevented tau-induced block of synaptic vesicle (SV) endocytosis and EPSC rundown.

in ppts increased to 22.6% ± 2.4%, indicating sequestration of dynamin 1 by MTs. When PHDP5 was added to MT and dynamin 1, dynamin 1 in the ppt fraction decreased to 6.3% ± 2.4%, indicating that PHDP5 works as a DN peptide for inhibiting MT-dynamin interactions (**Figure 5A**).

A cryo-electron microscope study on dynamin 1 assembled on lipid membrane has revealed that PH domain is tucked up into dynamin structure in apo state, but upon GTP binding, exposed toward membrane by a conformational change (**Kong et al., 2018**). In negatively stained electron micrographs, dynamin 1 is periodically arranged on the surface of MTs (**La et al., 2020**), suggesting a helical polymerization like in dynamin-membrane interaction (**Zhang and Hinshaw, 2001**). Therefore, PH domain including the putative-binding site PHDP5 is likely exposed toward MT surface. To examine whether PH domain of dynamin 1 can directly bind to MTs, immunofluorescence labeled MTs and glutathione transferase-tagged PH domain (GST-PH) were mixed and observed by confocal and electron microscopy (**Figure 5—figure supplement 2**). In confocal microscopic imaging, GST-PH co-localized with MTs, in contrast to controls, where MTs were mixed with GST alone (**Figure 5—figure supplement 2A**). These results were further confirmed in electron microscopic imaging, showing co-localizations of MTs and GST-PH (**Figure 5—figure supplement 2B**). Thus, dynamin 1 PH domain can associate with MTs, although it remains to be determined whether PHDP5 can directly bind to MTs.

Loading of PHDP5 (0.25 mM) alone in calyceal terminals had no effect on exo-endocytic capacitance changes, but when co-loaded with WT h-tau (20 µM), it significantly attenuated the h-tau-induced endocytic slowing (p < 0.05, **Figure 5B**). Scrambled PHDP5 peptide (0.25 mM) loaded as a control had no effect on h-tau-induced endocytic slowing. Like its effect on capacitance changes, intra-terminal infusion of PHDP5 alone (1 mM) did not affect EPSC amplitude, but when co-loaded with WT h-tau (20 µM), significantly attenuated the inhibitory effect of h-tau on EPSC amplitude (p < 0.01, **Figure 5C**). Co-infusion of scrambled PHDP5 (1 mM) with h-tau (20 µM) did not affect the h-tau-induced EPSC rundown (p = 0.46). These results further support that WT h-tau causes dynamin deficiency via new assembly of MTs thereby impairing SV endocytosis and synaptic transmission. These results also highlight PHDP5 as a potential therapeutic tool for rescuing synaptic dysfunctions associated with AD or Parkinson's disease (PD).

## Discussion

Using the calyx of Held in brainstem slices as an AD model for dissecting mammalian central excitatory synaptic transmission, we demonstrated that intra-terminal loading of WT h-tau impairs vesicle endocytosis and synaptic transmission via de novo MT assembly. Previous overexpression studies in cultured cells reported MT assembly by injection or overexpression of WT tau (**Thies and Mandelkow, 2007**; **Drubin and Kirschner, 1986**; **Shemesh et al., 2008**) or phosphorylated tau (**Shahpasand et al., 2012**; **Liu et al., 2007**). Compared with overexpression, our whole-cell method allows targeted loading of molecules in presynaptic terminals at defined concentrations because of a large pipette-to-cell volume ratio (**Pusch and Neher, 1988**). In postmortem brain tissue homogenates from AD patients, soluble tau content is estimated as 6 ng/µg of protein, which is eight times higher than controls (**Khatoon et al., 1992**). Assuming protein contents in brain homogenate as 10%, 60 kDa tau concentration in AD patients' brain is estimated as 10 µM. Since elevation of soluble tau concentration likely occurs mainly in axons and axon terminal compartments of neurons, soluble tau concentration in AD patients in presynaptic terminals can be higher than that. Our results at the calyx of Held suggest that excitatory synaptic transmission, in general, can be significantly impaired in such situations. In

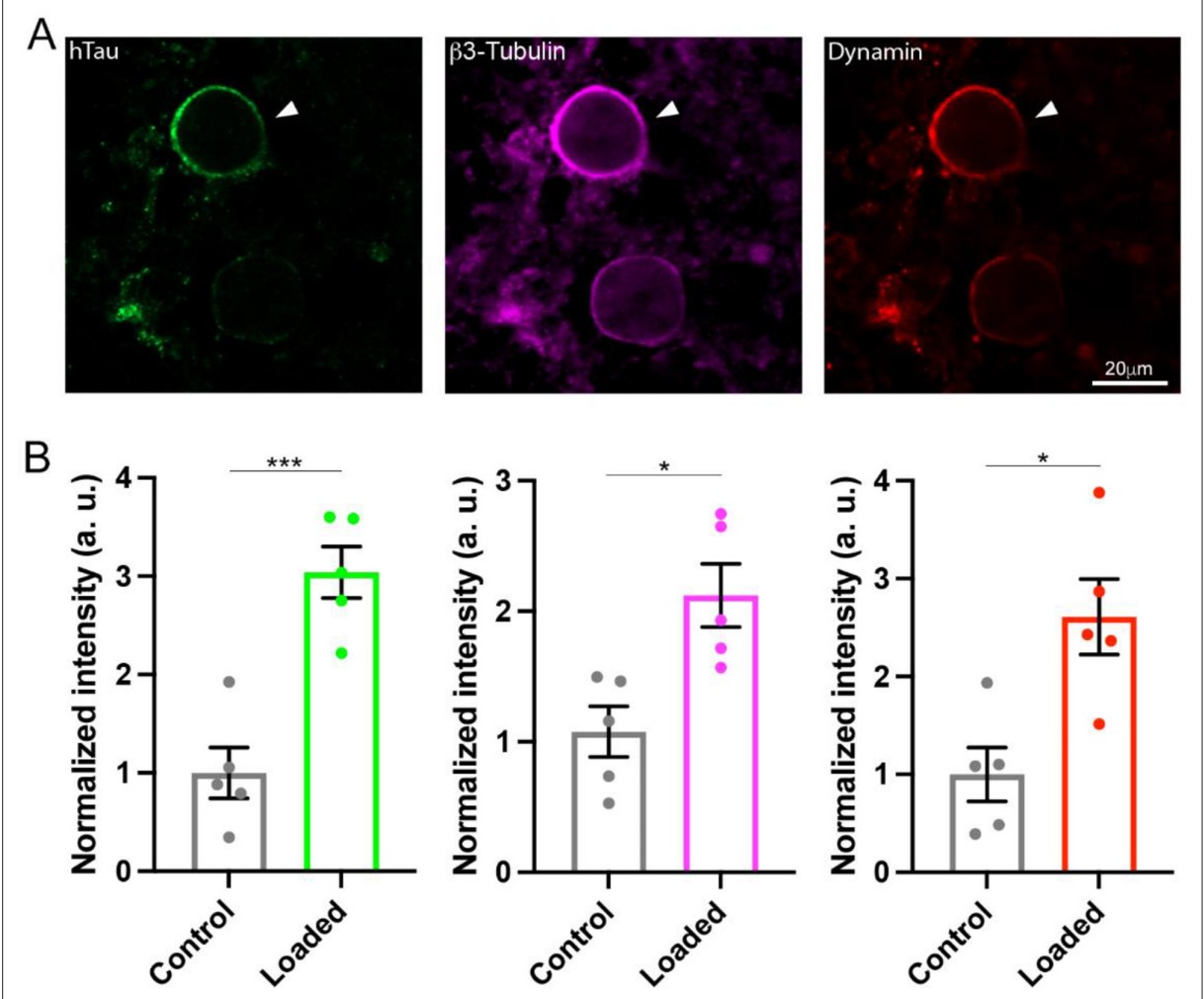

**Figure 4.** Wild-type (WT) human tau (h-tau) assembled microtubules (MTs) and increased bound-form dynamins in calyceal terminals.
(**A**) Immunofluorescence images of brainstem slices showing loaded WT h-tau (green, left, arrowhead) labeled with anti h-tau/Alexa Fluor 488 antibodies (green, left panel), newly assembled MTs labeled with anti-β3-tubulin/Alexa Fluor 647 antibodies (magenta, middle), and increased bound-form dynamin labeled with anti-dynamin 1/Alexa Fluor 568 antibodies (red, right panel). (**B**) Bar graphs showing immunofluorescence intensities of h-tau (green), β3-tubulin (magenta), and dynamin (red) relative to controls with no loading (black bars). WT h-tau loading significantly increased β3-tubulin (p = 0.0105) and dynamin 1 (p = 0.0109) intensity in terminals compared to control terminals without WT h-tau loading (n = 5 terminals from 5 slices for each data set, two-tailed unpaired t-test with Welch's correction; *p < 0.05, ***p < 0.001).

The online version of this article includes the following source data and figure supplement(s) for figure 4:

**Source data 1.** Raw immunofluorescence images from *Figure 4A*.

**Source data 2.** Data from *Figure 4B*.

**Figure supplement 1.** Super-resolution imaging of human tau (h-tau)-infused calyx of Held.

**Figure supplement 1—source data 1.** Raw images from *Figure 4—figure supplement 1*.

**Figure supplement 2.** Effects of a formin mDia inhibitor on synaptic vesicle (SV) endocytosis and tau infusion on the quantal EPSCs.

**Figure supplement 2—source data 1.** Data from *Figure 4—figure supplement 2*.

fact, the magnitude of EPSC rundown after WT h-tau loading is comparable to that caused by the clinical dose of general anesthetic isoflurane at the calyx of Held in slice (*Wang et al., 2020*). In AD, tau pathology starts from the locus coeruleus in the brainstem and undergoes trans-synaptic propagation to hippocampal and neocortical neurons (*Goedert et al., 2014*). Present results in our model synapse suggest that synaptic functions in such tau-propagation pathways can be severely affected at the early stage of AD.

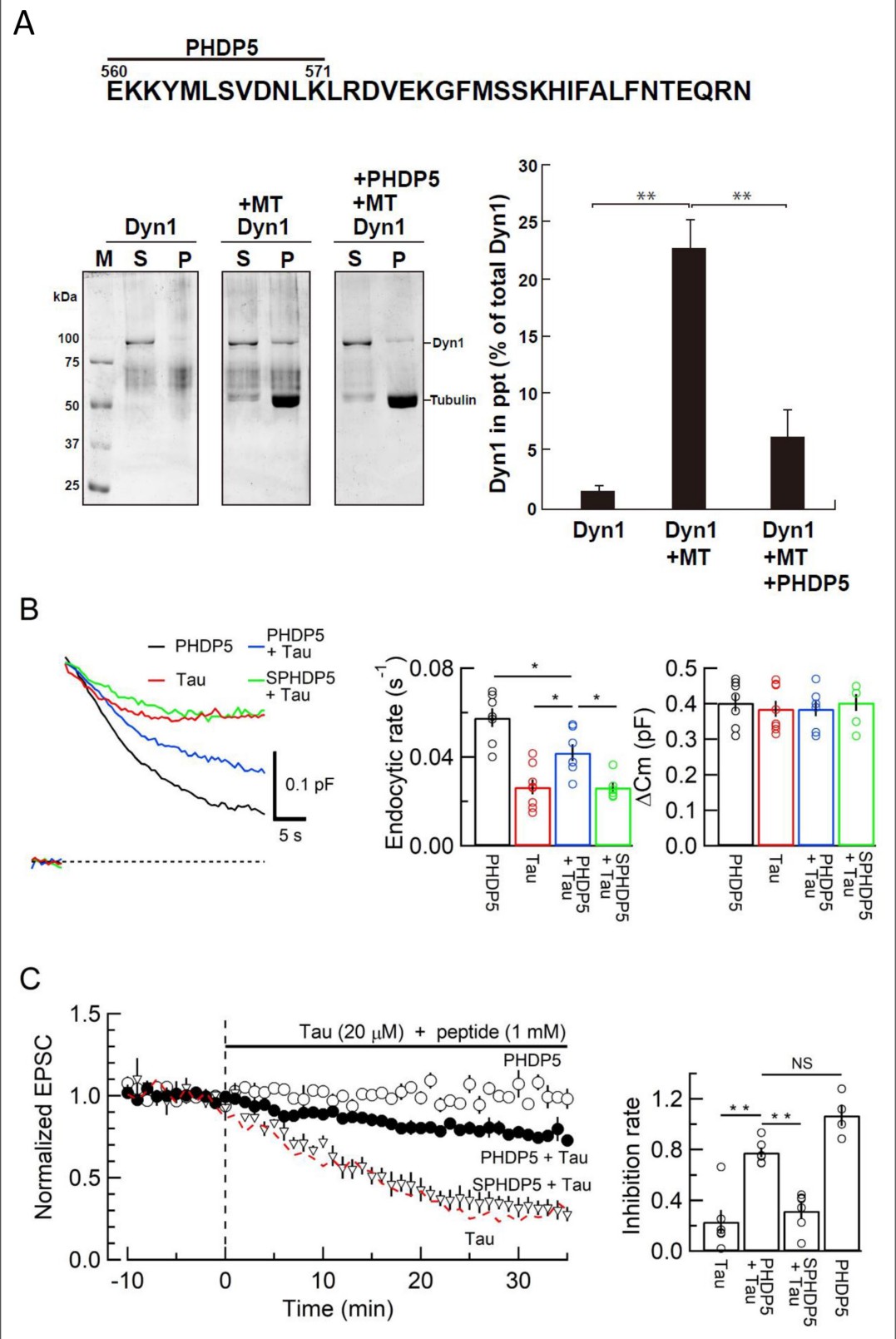

**Figure 5.** Dynamin 1 pleckstrin-homology (PH) domain peptide inhibited microtubule (MT)-dynamin 1 binding and prevented endocytic slowing and EPSC rundown caused by wild-type (WT) human tau (h-tau). (**A**) *Top*, partial amino acid sequence of PH domain of mouse dynamin 1 indicating the sequence of the synthetic dodecapeptide PHDP5 (560–571). *Left*, SDS-PAGE of MT-dynamin 1-binding assay. S, supernatant; P, precipitates. Dyn1, dynamin

*Figure 5 continued on next page*

*Figure 5 continued*

1. *Right*, quantification of MT-dynamin 1 interaction. The bars indicate the percentage of dynamin 1 found in precipitates relative to total amount. PHDP5 significantly inhibited MT-dynamin 1 interaction (**p < 0.01, n = 3). (**B**) Presynaptic membrane capacitance records (superimposed) after loading h-tau alone (20 μM, red trace, taken from *Figure 4B*), h-tau co-loaded with DPHP5 (0.25 mM, blue) or scrambled DPHP5 (SDPHP5, green). DPHP5 alone (0.25 mM, black trace, 7 terminals from 7 slices) had no effect on capacitance changes compared to non-loading terminal controls (taken from *Figure 3B*). Bar graphs of endocytic rates (middle panel) indicate significant difference (*p < 0.05, 7 terminals from 7 slices) between tau (red bar, 8 terminals from 8 slices) and DPHP5 + tau (blue, 7 terminals from 7 slices) as well as between SDPHP5 + tau (blue) and DPHP5 + tau (green, n = 6 terminals from 6 slices). The magnitudes of exocytic capacitance changes were not significantly different between the groups, recorded 25 min after rupture. (**C**) DPHP5 attenuated h-tau-induced EPSC rundown. The EPSC rundown after h-tau infusion (20 μM, red dashed line; data taken from *Figure 1A*) was attenuated by DPHP5 (1 mM) co-loaded with h-tau (filled circles) but not by scrambled DPHP5 peptide (SDPHP5, open triangles, 1 mM). DPHP5 alone (1 mM, open circles) had no effect on EPSC amplitude throughout. Bar graphs indicate EPSC amplitude (normalized to that before infusion) 30 min after infusion. Significant difference (**p < 0.01) between tau and tau + DPHP5, between tau + DPHP5 and tau + SDPHP5. The difference between DPHP5 alone and DPHP5 + tau was not significant (p = 0.09), indicating the partial antagonistic effect of DPHP5 against h-tau-induced EPSC rundown.

The online version of this article includes the following source data and figure supplement(s) for figure 5:

**Source data 1.** Data from *Figure 5A*.

**Source data 2.** Data from *Figure 5B and C*.

**Source data 3.** Images from *Figure 5A*.

**Figure supplement 1.** PHDP5 strongly inhibited dynamin 1 binding to microtubules (MTs).

**Figure supplement 1—source data 1.** Images from *Figure 5—figure supplement 1*.

**Figure supplement 1—source data 2.** Data from *Figure 5—figure supplement 1*.

**Figure supplement 2.** Pleckstrin-homology (PH) domain of dynamin 1 directly binds to microtubules.

**Figure supplement 2—source data 1.** Images from *Figure 5—figure supplement 2*.

---

Membrane capacitance measurements at the calyx of Held revealed the primary target of WT h-tau toxicity as SV endocytosis. Endocytic slowing impairs SV recycling and reuse, thereby inhibiting SV exocytosis, particularly in response to high-frequency stimulations (*Yamashita et al., 2005*). The toxic effects of h-tau on SV endocytosis and synaptic transmission were prevented by nocodazole co-application. Together with the lack of toxicity of del-MTBD and toxic effects of taxol on synaptic transmission, these results suggest pathological roles of over-assembled MTs. Like WT h-tau, intra-terminal loading of WT α-synuclein slows SV endocytosis and impairs fidelity of high-frequency neurotransmission at the calyx of Held (*Eguchi et al., 2017*). α-Synuclein toxicities can be rescued by blocking MT assembly with nocodazole or a photosensitive colchicine derivative PST-1. Thus, a common mechanism likely underlies synaptic dysfunctions in AD and PD. Compared with α-synuclein, h-tau toxicity is much stronger on endocytosis as well as on synaptic transmission. Thus, abnormal elevation of endogenous molecules beyond homeostatic level may cause AD and PD symptoms, like many other human diseases.

Although the GTPase dynamin is a well-known player in endocytic fission of SVs (*Hinshaw and Schmid, 1995*; *Takei et al., 1995*), it was originally discovered as an MT-binding protein (*Shpetner and Vallee, 1989*). Subsequent studies indicated that this interaction upregulates dynamin's GTPase activity (*Maeda et al., 1992*; *Shpetner and Vallee, 1992*) and can induce MT instability with dynamin 2 (*Tanabe and Takei, 2009*) or stabilizes MT bundle formation with dynamin 1 (*La et al., 2020*). However, the binding domain of dynamin remained unidentified. In this study, calyceal terminals loaded with WT h-tau showed a prominent increase in immunofluorescence signal intensity corresponding to bound dynamins. This was associated with an elevation in intra-terminal MTs, suggesting that newly assembled MTs induced by loaded h-tau sequestered cytosolic dynamins. These results well explain impairments of SV endocytosis by intra-terminal h-tau loading. Through synthetic peptide screening, we found that a dodecapeptide from dynamin 1 PH domain significantly inhibited the MT-dynamin interaction. This peptide PHDP5 is ~80% homologous to dynamin 3, another isoform involved in vesicle endocytosis (*Raimondi et al., 2011*). Although direct binding of this peptide to MTs remains to be seen, it significantly rescued endocytic impairments and EPSC rundown induced by intra-terminal WT h-tau. Hence, MTs over-assembled by soluble WT h-tau proteins likely sequester

free dynamins in presynaptic terminals, thereby blocking SV endocytosis and synaptic transmission, at least in this slice model. This dynamin sequestration mechanism by newly assembled MTs may also underlie the toxic effect of α-synuclein on SV endocytosis (*Eguchi et al., 2017*) in PD.

Unlike WT h-tau, the FTDP-linked mutant tau does not affect SV endocytosis (*Zhou et al., 2017*), but binds to both actin filaments (*Fulga et al., 2007*) and the SV transmembrane protein synaptogyrin (*McInnes et al., 2018*), thereby immobilizing SVs (*McInnes et al., 2018*; *Zhou et al., 2017*). WT-tau can also bind to synaptogyrin (*McInnes et al., 2018*), but cannot bind to F-actins because of a difference in the MT-binding regions between FTDP mutant and WT tau (*Drubin and Kirschner, 1986*; *Roger et al., 2004*). However, WT h-tau can bind to various other macromolecules and organelles such as MTs, neurofilaments, and ribosomes (*Guo et al., 2017*) as well as to synaptogyrin, thereby possibly immobilizing SVs. Recycling transport of SVs impaired by this mechanism might additionally contribute to the rundown of synaptic transmission remaining unblocked by the MT-dynamin blocker peptide.

In the absence of a powerful tool for alleviating symptoms associated with AD or PD, the calyx of Held slice model might provide a platform upon which therapeutic tools for rescuing synaptic dysfunctions can be pursued. The combination of this slice model with animal models could provide a new pathway toward rescuing neurological disorders.

## Materials and methods

### Animals

All experiments were performed in accordance with the guidelines of the Physiological Society of Japan and animal experiment regulations at Okinawa Institute of Science and Technology Graduate University.

### Recombinant human tau preparation

Human tau (h-tau) lacking the MT-binding domain (amino acid 244–367, del-MTBD) were produced by site-directed mutagenesis as previously reported (*Xie et al., 2014*). Wild-type (WT) and del-MTBD mutant h-tau of 0N4R isoform were expressed in *E. coli.* (BL21/DE3) and purified as described previously (*Hasegawa et al., 1998*) with minor modifications. Briefly, harvested bacteria expressing recombinant tau were lysed in homogenization buffer (50 mM PIPES, 1 mM EGTA, 1 mM DTT, 0.5 mM PMSF, and 5 µg/ml Leupeptin, pH6.4), sonicated and centrifuged at $27,000 \times g$ for 15 min. Supernatants were charged onto phosphocellulose column (P11, Whatman). After washing with homogenization buffer containing 0.1 M NaCl, h-tau-containing fractions were eluted by the buffer containing 0.3 M NaCl. Subsequently, the proteins were precipitated by 50% saturated ammonium sulfate and re-solubilized in homogenization buffer containing 0.5 M NaCl and 1% 2-mercaptoethanol. After incubation at 100°C for 5 min, heatstable (soluble) fractions were obtained by centrifugation at $21,900 \times g$, and fractionated by reverse phase high-performance liquid chromatography (RP-HPLC) using Cosmosyl Protein-R (Nacalai tesque Inc). Aliquots of h-tau containing fractions were lyophilized and stored at –80°C. Purified h-tau proteins were quantified by SDS-PAGE followed by Coomassie Brilliant Blue staining.

### Purification of recombinant human dynamin 1 protein

His-tagged human dynamin 1 was expressed using the Bac-to-Bac baculovirus expression system (Thermo Fisher Scientific, Waltham, MA) and purified as described previously (*Yamada et al., 2013*). The purified dynamin solutions were concentrated using Centriplus YM50 (cat#4310; Merck-Millipore, Darmstadt, Germany).

### MT polymerization assay

Effects of tau and nocodazole on MT polymerization were tested using a Tubulin Polymerization Assay (Cytoskeleton Inc, Denver, CO). Briefly, purified WT or del-MTBD mutant h-tau (10 µM) were mixed with porcine tubulin (20 µM) in an assembly buffer at 37°C. Nocodazole was added to the mixture at 0 min of incubation. MT polymerization was fluorometrically assayed (excitation at 360 nm, emission at 465 nm) using Infinit F-200 Microplate Reader (TECAN, Männedorf/Switzerland) at 1 min intervals for 30 min. After incubation, resultant solutions were subjected to centrifugation at $100,000 \times g$ for 15 min

at 20°C. Supernatants (free tubulin fraction) and pellets (MT fraction) were subjected to SDS-PAGE to quantify the amount of tubulin assembled into MTs.

## Peptide synthesis and LC-MS/MS analysis

The peptides were synthesized through conventional 9-fluorenylmethyloxycarbonyl (Fmoc) solid-phase peptide synthesis, onto preloaded Fmoc-alanine TCP-resins (Intavis Bioanalytical Instruments) using automated peptide synthesizer ResPep SL (Intavis Bioanalytical Instruments). All Fmoc-amino acids were purchased from Watanabe Chemical Industries and prepared at 0.5 M in *N*-methyl pyrrolidone (Wako Pure Chemical Industries). After synthesis, peptides were cleaved with (v/v/v) 92.5% TFA, 5% TIPS, and 2.5% water for 2 hr, precipitated using *tert*-butyl-methyl-ether at –30°C, pelleted, and resuspended in water before lyophilization (EYELA FDS-1000) overnight. All synthesized peptides' purity and sequence were then confirmed by LC-MS/MS using a Q-Exactive Plus Orbitrap hybrid mass spectrometer (Thermo Scientific) equipped with Ultimate 3000 nano-HPLC system (Dionex), HTC-PAL autosampler (CTC Analytics), and nanoelectrospray ion source.

## MT-dynamin-binding assay

MT-binding assay was performed using Microtubule Binding Protein Spin Down Assay Kit (cat#BK029, Cytoskeleton Inc, Denver, CO). Briefly, 20 µl of 5 mg/ml tubulin in general tubulin buffer (GTB; 80 mM PIPES pH 7.0, 2 mM $MgCl_2$, 0.5 mM EGTA) supplemented with 1 mM GTP were polymerized by adding 2 µl of cushion buffer (80 mM PIPES pH 7.0, 1 mM $MgCl_2$, 1 mM EGTA, 60% glycerol) and incubated at 35°C for 20 min. MTs were stabilized with 20 µM Taxol. Taxol-stabilized MTs (2.5 µM) and dynamin 1 (1 µM) were incubated in GTB with or without 1 mM peptide at RT for 30 min. After incubation, the 50 µl of mixture was loaded on top of 100 µl cushion buffer supplemented with 20 µM Taxol, and then centrifuged at 100,000× *g* for 40 min at RT. After the ultracentrifugation, 50 µl of supernatant was taken and mixed with 10 µl of 5× sample buffer. The resultant pellet was resuspended with 50 µl of 1× sample buffer. Twenty µl of each sample was analyzed by SDS-PAGE and stained with SYPRO Orange. Protein bands were visualized using FLA-3000 (FUJIFILM Co. LTD, Tokyo, Japan).

## Immunocytochemical analysis

The following primary antibodies were used: anti-β3-tubulin (Synaptic System, #302304), anti-h-tau (BioLegend, #806501), anti-dynamin (Invitrogen, PA1-660). Secondary antibodies were goat IgG conjugated with Alexa Fluor 488, 568, or 647 (Thermo Fisher Scientific). Acute brainstem slices (175 µm in thickness, see below) were fixed with 4% paraformaldehyde in PBS for 30 min at 37°C and overnight at 4°C. On the following day, slices were rinsed three times in PBS, permeabilized in PBS containing 0.5% Triton X-100 (Tx-100; Nacalai Tesque) for 30 min and blocked in PBS containing 3% bovine serum albumin (BSA; Sigma-Aldrich) and 0.05% Tx-100 for 45 min. Slices were incubated overnight at 4°C with primary antibody diluted in PBS 0.05% Tx-100, 0.3% BSA. On the next day, slices were rinsed three times with PBS containing 0.05% Tx-100 for 10 min and incubated with corresponding secondary antibody diluted in PBS 0.05% Tx-100, 0.3% BSA for 1 hr at RT. Slices were further rinsed three times in PBS 0.05% Tx-100 for 10 min and finally washed in PBS for another 10 min. Finally, slices were mounted on glass slides (Matsunami) using liquid mounting medium (Ibidi) and sealed using nail polish. Confocal images were acquired on laser scanning microscopes (LSM780 or LSM900, Carl Zeiss) equipped with a Plan-apochromat 63× oil immersion objective (1.4 NA) and 488, 561, and 633 nm excitation laser lines. For quantifying fluorescence intensity levels, the region of interest was delimited around calyceal terminals, and background fluorescence was subtracted using ImageJ software.

## Purification of GST-proteins

The cDNA encoding PH domain (521–618 amino acids) of human dynamin 1 (NM_004408.4) (*Gu et al., 2010*) were prepared by PCR and subcloned into the plasmid pGEX-6P vector. The resulting plasmid was transformed into bacterial BL21(DE3) pLysS strain for protein expression. The expression of GST-fusion proteins was induced by 0.1 mM isopropyl-1-thio-D-galactopyranoside at 37°C for 3–6 hr in LB media supplemented with 100 µg/ml ampicillin at $A_{600}$ = 0.8. GST-fusion proteins were then purified as described (*Slepnev et al., 2000*). The nucleotide sequences of the constructs used in this study were verified with DNA sequence analysis. All the purified protein solutions (1–3 mg/ml) were stored at –80°C and thawed at 37°C before use.

## Microscopic observation of MT and GST-PH protein

GST or GST-PH was labeled using HiLyte Fluor-555 labeling kit according to manufacturer's manual (cat#LK14, Dojindo Co. LTD, Kumamoto, Japan). HiLyte Fluor-555 labeled GST or GST-PH was mixed with non-labeled each protein at the ratio of 1:1.2. Flutax1-stabilized MTs (4.1 µM) and fluorescent GST or GST-PH at 11 µM were mixed in GTB containing 2 µM Flutax1 at 37°C for 60 min. Eight µl of the mixture was spotted on the slide glass and mounted with Fluoromount (cat#K024, Diagnostic BioSystems, Pleasanton, CA). Samples were examined using a spinning disc confocal microscope system (X-Light Confocal Imager; CREST OPTICS S.P.A., Rome, Italy) combined with an inverted microscope (IX-71; Olympus Optical Co., Ltd., Tokyo, Japan) and an iXon+ camera (Oxford Instruments, Oxfordshire, UK). The confocal system was controlled by MetaMorph software (Molecular Devices, Sunnyvale, CA). When necessary, images were processed using Adobe Photoshop CS3 or Illustrator CS3 software. For electron microscopic observation, samples were submitted to negative staining for imaging with a transmission electron microscope (H-7650, Hitachi High-Tech Corp., Tokyo, Japan) at 120 kV.

## Slice electrophysiology

After killing C57BL/6N mice of either sex (P13–15) by decapitation under isoflurane anesthesia, brain-stems were isolated and transverse slices (175 µm thick) containing the medial nucleus of the trapezoid body (MNTB) were cut using a vibratome (VT1200S, Leica) in ice-cold artificial cerebrospinal fluid (aCSF, see below) with reduced $Ca^{2+}$ (0.1 mM) and increased $Mg^{2+}$ (3 mM) concentrations or sucrose-based aCSF (NaCl was replaced to 300 mM sucrose, concentrations of $CaCl_2$ and $MgCl_2$ was 0.1 and 6 mM, respectively). Slices were incubated for 1 hr at 36–37°C in standard aCSF containing (in mM): 125 NaCl, 2.5 KCl, 26 $NaHCO_3$, 1.25 $NaHPO_4$, 2 $CaCl_2$, 1 $MgCl_2$, 10 glucose, 3 myo-inositol, 2 sodium pyruvate, and 0.5 sodium ascorbate (pH 7.4 when bubbled with 95% $O_2$ and 5% $CO_2$, 310–320 mOsm), and maintained thereafter at RT (24–28°C).

Whole-cell recordings were made using a patch-clamp amplifier (Multiclamp 700A, Molecular Devices, Sunnyvale, CA, for pair recordings and EPC-10 USB, HEKA Elektronik, Germany, for presynaptic capacitance measurements) from the calyx of Held presynaptic terminals and postsynaptic MNTB principal neurons visually identified with a 60× or 40× water immersion objective (LUMPlanFL, Olympus) attached to an upright microscope (Axioskop2, Carl Zeiss, or BX51WI, Olympus, Japan). Data were acquired at a sampling rate of 50 kHz using pClamp (for Multiclamp 700A) or Patchmaster software (for EPC-10 USB) after online filtering at 5 kHz. The presynaptic pipette was pulled for the resistance of 7–10 MΩ and had a series resistance of 14–20 MΩ, which was compensated by 70% for its final value to be 7 MΩ. Resistance of the postsynaptic pipette was 5–7 MΩ, and its series resistance was 10–25 MΩ, which was compensated by up to 75% to a final value of 7 MΩ. The aCSF routinely contained picrotoxin (10 µM) and strychnine hydrochloride (0.5 µM) to block $GABA_A$ receptors and glycine receptors, respectively. Postsynaptic pipette solution contained (in mM): 130 CsCl, 5 EGTA, 1 $MgCl_2$, 5 QX314-Cl, 10 HEPES (adjusted to pH 7.3–7.4 with CsOH). The presynaptic pipette solution contained (in mM): 105 K methanesulfonate, 30 KCl, 40 HEPES, 0.5 EGTA, 1 $MgCl_2$, 12 phosphocreatine (Na salt), 3 ATP (Mg salt), 0.3 GTP (Na salt) (pH 7.3–7.4 adjusted with KOH, 315–320 mOsm).

In simultaneous presynaptic and postsynaptic whole-cell recordings, postsynaptic MNTB neurons were voltage-clamped at the holding potential of –70 mV, and EPSCs were evoked, at 0.1 or 1 Hz, by action potentials elicited by a depolarizing current (1 ms) injected in calyceal terminals. For intra-terminal loading of taxol (1 µM), it was diluted in presynaptic pipette solution from 5 mM DMSO stock for final DMSO concentration to be 0.02%. Likewise, nocodazole (20 µM, 0.1% DMSO) was included in presynaptic pipette solution. Presynaptic pipette solutions in nocodazole controls contained 0.1% DMSO. In simultaneous pre- and postsynaptic recordings, WT h-tau, del-MTBD tau, taxol, or synthetic peptides were loaded in calyceal terminals using the pipette perfusion technique (*Hori et al., 1999*; *Hori and Takahashi, 2012*). Briefly, a fine superfusion tube composed of plastic and glass tubes was installed in a presynaptic patch pipette. After backfilling the tube with pipette solutions containing proteins and/or peptides, it was inserted into a patch pipette with its tip 500–600 µm behind the tip of presynaptic patch pipette. After recording baseline EPSCs, the tube solution was delivered into presynaptic patch pipette with a positive pressure (8–10 psi) applied using a pico-pump.

Membrane capacitance ($C_m$) measurements were made from calyx of Held presynaptic terminals in the whole-cell configuration at RT (*Sun and Wu, 2001*; *Yamashita et al., 2005*). Calyceal terminals

were voltage-clamped at a holding potential of –80 mV, and a sinusoidal voltage command (1 kHz, 60 mV in peak-to-peak amplitude) was applied. To isolate presynaptic voltage-gated $Ca^{2+}$ currents ($I_{Ca}$), the aCSF contained 10 mM tetraethylammonium chloride, 0.5 mM 4-aminopyridine, 1 µM tetrodotoxin, 10 µM bicuculline methiodide, and 0.5 µM strychnine hydrochloride. The presynaptic pipette solution contained (in mM): 125 Cs methanesulfonate, 30 CsCl, 10 HEPES, 0.5 EGTA, 12 disodium phosphocreatine, 3 MgATP, 1 $MgCl_2$, 0.3 $Na_2GTP$ (pH 7.3 adjusted with CsOH, 315–320 mOsm). Tau or synthetic peptides were dissolved in pipette solution and backfilled into the pipette briefly after loading the tau-free pipette solution from the pipette tip. Care was taken to maintain series resistance <16 MΩ to allow dialysis of the terminal with pipette solution. Recording pipette tips were coated with dental wax to minimize stray capacitance (4–5 pF). Single square pulse (–80 to 10 mV, 20 ms duration) was used to induce presynaptic $I_{Ca}$. In these experiments, exocytic capacitance change ($\Delta C_m$) represents ~5 times larger number of SVs (estimated from $\Delta C_m$ divided by $C_m$ of single SV) than that in the immediately releasable pool by presynaptic action potentials (estimated by the size of maximally evoked EPSCs divided by the size of miniature EPSCs). Membrane capacitance changes within 450 ms of square-pulse stimulation were excluded from analysis to avoid contamination by conductance-dependent capacitance artifacts (*Yamashita et al., 2005*). To avoid the influence of capacitance drift on baseline, we removed data when the baseline drift measured 0–10 s before stimulation was over 5 fF/s. When the drift was 1–5 fF/s, we subtracted a linear regression line of the baseline from the data for the baseline correction. The endocytic rate was calculated from the slope of the normalized $C_m$ changes during the initial 10 s after the stimulation.

## Data analysis and statistics

Data were analyzed using IGOR Pro 6 (WaveMatrics), Excel 2016 (Microsoft), and StatPlus (AnalystSoft Inc) and KaleidaGraph for Macintosh, version 4.1 (Synergy Software Inc, Essex Junction, VT). All values are given as mean ± SEM. Differences were considered statistically significant at $p < 0.05$ in paired or unpaired t-tests, one-way ANOVA with Scheffe post hoc test and repeated-measures two-way ANOVA with post hoc Scheffe test.

## Study approval

All experiments were carried out in accordance with the regulation and guidelines of Okinawa Institute of Science and Technology Graduate University (Approval number: 2021-347-00).

## Acknowledgements

We thank Yasuo Ihara, Nobuyuki Nukina, and Takeshi Sakaba for comments and Patrick Stoney for editing this paper. We also thank Shota Okuda and Mikako Matsubara for their contributions in the early stage of this study, and Satoko Wada-Kakuda for technical assistant with in vitro analysis of tau. This research was supported by funding from Okinawa Institute of Science and Technology and from Technology (OIST) and Core Research for the Evolutional Science and Technology of Japan Science and Technology Agency (CREST) to TT, and by Scientific Research on Innovative Areas to TM (Brain Protein Aging and Dementia Control 26117004).

## Additional information

### Funding

| Funder | Grant reference number | Author |
| --- | --- | --- |
| Okinawa Institute of Science and Technology Graduate University | | Tomoyuki Takahashi |
| Core Research for Evolutional Science and Technology | | Tomoyuki Takahashi |

| Funder | Grant reference number | Author |
|---|---|---|
| Japan Society for the Promotion of Science | Grant-in-Aid for Scientific Research on Innovative Areas | Tomohiro Miyasaka |
| Japan Society for the Promotion of Science | 26117004 | Tomohiro Miyasaka |

The funders had no role in study design, data collection and interpretation, or the decision to submit the work for publication.

## Author contributions

Tetsuya Hori, Conceptualization, Formal analysis, Investigation, Methodology, Project administration, Validation, Visualization, Writing - original draft, Writing - review and editing; Kohgaku Eguchi, Han-Ying Wang, Satyajit Mahapatra, Formal analysis, Investigation, Visualization, Writing - review and editing; Tomohiro Miyasaka, Formal analysis, Funding acquisition, Investigation, Resources, Visualization, Writing - review and editing; Laurent Guillaud, Formal analysis, Investigation, Methodology, Visualization, Writing - review and editing; Zacharie Taoufiq, Formal analysis, Investigation, Resources, Visualization, Writing - review and editing; Hiroshi Yamada, Formal analysis, Investigation, Project administration, Resources, Supervision, Visualization, Writing - review and editing; Kohji Takei, Conceptualization, Investigation, Project administration, Resources, Supervision, Visualization, Writing - original draft, Writing - review and editing; Tomoyuki Takahashi, Conceptualization, Data curation, Project administration, Supervision, Visualization, Writing - original draft, Writing - review and editing

## Author ORCIDs

Tetsuya Hori ⓘ http://orcid.org/0000-0002-0823-3306
Kohgaku Eguchi ⓘ http://orcid.org/0000-0002-6170-2546
Han-Ying Wang ⓘ http://orcid.org/0000-0002-3021-7134
Laurent Guillaud ⓘ http://orcid.org/0000-0002-9688-0991
Tomoyuki Takahashi ⓘ http://orcid.org/0000-0002-8771-7666

## Ethics

All experiments were performed in accordance with the guidelines of the Physiological Society of Japan and animal experiment regulations at Okinawa Institute of Science and Technology Graduate University.

## Decision letter and Author response

Decision letter https://doi.org/10.7554/eLife.73542.sa1
Author response https://doi.org/10.7554/eLife.73542.sa2

---

# Additional files

## Supplementary files

• Transparent reporting form

## Data availability

All data generated or analysed during this study are included in the manuscript and supporting file; Source Data files have been provided for All Figures.

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
