## [Editor Report]

This study provides an interesting new insight into the synaptic disease mechanisms of tauopathies. The paper is based on a technically very rigorous dataset indicating that increased levels of soluble Tau impair pre-synaptic endocytosis and, consequently, neurotransmission by sequestering Dynamin-1 on microtubules. The findings are of major relevance for basic neuronal cell biology and translational neuroscience alike.

---

## [Decision Letter]

**Decision letter after peer review:**

Thank you for submitting your article "Microtubule assembly by soluble tau impairs vesicle endocytosis and excitatory neurotransmission via dynamin sequestration in Alzheimer's disease synapse model" for consideration by *eLife*. Your article has been reviewed by 2 peer reviewers, and the evaluation has been overseen by a Reviewing Editor and Suzanne Pfeffer as the Senior Editor. The reviewers have opted to remain anonymous.

The reviewers have discussed their reviews with one another, and the Reviewing Editor has drafted the comments below to help you prepare a revised submission.

Essential revisions - Additional Experiments:

1. The PH domain-derived peptide used for the experiments documented in Figure 5 is part of a tightly folded domain (i.e. as a loop-β-fold, see PDB 2YS1). No data are presented to demonstrate that the PH domain of Dynamin-1 indeed binds to microtubules directly (or indirectly) in vitro and in vivo (or in living neurons). Indeed, it is questionable that the short PHDP5 12-mer peptide, when presented outside its sequence context within the PH domain folds into its native structure and exhibits its physiological set of interactions. A thorough biochemical characterization of the association of Dynamin-1 with microtubules is a prerequisite to interpret the results shown in Figure 5, which indicate a partial rescue of Tau induced endocytic defects by a rather high concentration (1 mM) of PHDP5 peptide. Moreover, additional complementary tools (e.g. expression or injection of WT vs. mutant forms of Dynamin-1 or its PH domain) should be used to assess the role of the alleged microtubule binding site in Dynamin-1 that is proposed to mediate the effects of Tau on pre-synaptic function. These issues need to be addressed experimentally.

2. The link between elevated microtubule assembly and Dynamin-1 recruitment discussed in the context of Figure 4 remains loose. In the confocal images of fluorescently labeled calyces in Figure 4A, beta3-Tubulin and Dynamin-1 are shown. Is there direct evidence to show that the alleged increase in beta3-Tubulin levels is due to elevated microtubule assembly or stability elicited by microinjected tau? Confocal resolution is insufficient to differentiate Dynamin-1 located in the pre-synaptic cytoplasm from Dynamin-1 bound to microtubules. Super-resolution imaging is required to show directly that Dynamin-1 accumulates on microtubules of Tau-injected calyces. Furthermore, to demonstrate that the Tau-microtubule-Dynamin-1 link is specifically at the core of the deleterious effects of elevated Tau levels, it needs to be shown that actin organization is unperturbed as expected, and that the levels of key pre-synaptic proteins (e.g. endophilin) and their sub-synaptic distribution are unaffected by Tau. These issues need to be addressed experimentally.

Essential Revisions - Optional Additional Experiments or Text Redaction

The reviewers identified two additional important issues that need to be addressed. If corresponding data are available or can be generated rapidly during the revision phase, they should be included. If this turns out to be problematic, the issues should be covered in the discussion part of the paper - e.g. as an outlook (points 3 and 4) or in the context of defining the general relevance and limitations of the present study (point 4).

3. It would be very interesting to assess the effects of elevated Tau levels on other, better characterized microtubule binding proteins in the pre-synapse, e.g. Formins such as mDia, which associate with actin and microtubules and have been implicated in pre-synaptic endocytosis. Moreover, evidence from neuroendocrine cells indicates that microtubules serve as platforms to direct exo-endocytic coupling by promoting 'hopping' of Clathrin structures, so that the question arises as to whether contributions of such processes to the observed pre-synaptic rundown and endocytic blockade can be ruled out.

4. Given the prominent cortical pathology of Alzheimer's Disease, it would be very informative to assess whether the principal observation that increased levels of soluble Tau sequester Dynamin-1 on microtubules holds true for synapses other than the Calyx of Held, for example central cortical synapses.

---

## [Author Response]

Essential revisions - Additional Experiments:1. The PH domain-derived peptide used for the experiments documented in Figure 5 is part of a tightly folded domain (i.e. as a loop-β-fold, see PDB 2YS1). No data are presented to demonstrate that the PH domain of Dynamin-1 indeed binds to microtubules directly (or indirectly) in vitro and in vivo (or in living neurons). Indeed, it is questionable that the short PHDP5 12-mer peptide, when presented outside its sequence context within the PH domain folds into its native structure and exhibits its physiological set of interactions.

We performed confocal and electron microscopic imaging using purified MT and GST-tagged PH domain of dynamin 1 (GST-PH). The results strongly suggest that the dynamin 1 PH domain can directly interact with MTs (new Suppl Figure S4, explanations in the text p5, third paragraph).

As reported in cryo-EM study on dynamin assembled on lipid membrane (Konig et al., 2018 Nature), upon GTP hydrolysis the PH domain can be exposed toward membrane by a conformational change from a position tucked up into dynamin structure. Like the dynamin-lipid membrane interaction (Zhang and Hinshaw, 2001 Nat Cell Biol), dynamin-MT interaction involves periodical arrangements of dynamins on the MT surface, suggesting their helical polymerization (La et al., 2020 FASEB J). Therefore, dynamin 1 PH domain comprising the putative binding site PHDP5 can be exposed toward MT surface (p5, third paragraph).

A thorough biochemical characterization of the association of Dynamin-1 with microtubules is a prerequisite to interpret the results shown in Figure 5, which indicate a partial rescue of Tau induced endocytic defects by a rather high concentration (1 mM) of PHDP5 peptide. Moreover, additional complementary tools (e.g. expression or injection of WT vs. mutant forms of Dynamin-1 or its PH domain) should be used to assess the role of the alleged microtubule binding site in Dynamin-1 that is proposed to mediate the effects of Tau on pre-synaptic function. These issues need to be addressed experimentally.

We performed biochemical experiments trying to characterize binding of dynamin 1 PH domain to microtubules. Unfortunately, however, GST-PH formed large complexes (Figure S4B) and sedimented into pellet after centrifugation, therefore we were unable to separate MTs between bound and unbound forms. We then performed glutathione beads separation experiments, but it did not work since MT bound to the beads even without PH domain. We did not perform overexpression experiments of PH domain-deleted/mutated dynamin 1 since it can change dynamin 3D structure. The concentrations of PHDP5 (0.25-1.0 mM) used for intra-terminal injection is comparable to that of dynamin 1 PRD peptide (1 mM) shown to block vesicle endocytosis by intra-terminal loading at the calyx of Held (Yamashita et al., 2005 Science).

2. The link between elevated microtubule assembly and Dynamin-1 recruitment discussed in the context of Figure 4 remains loose. In the confocal images of fluorescently labeled calyces in Figure 4A, beta3-Tubulin and Dynamin-1 are shown. Is there direct evidence to show that the alleged increase in beta3-Tubulin levels is due to elevated microtubule assembly or stability elicited by microinjected tau? Confocal resolution is insufficient to differentiate Dynamin-1 located in the pre-synaptic cytoplasm from Dynamin-1 bound to microtubules. Super-resolution imaging is required to show directly that Dynamin-1 accumulates on microtubules of Tau-injected calyces.

Although presynaptic terminals tend to suffer from whole-cell perturbation and pipette retraction after tau loading, we managed to have obtained super-resolution imaging data showing that dynamins are colocalized with microtubules in a tau-infused presynaptic terminal (Figure S2).

Furthermore, to demonstrate that the Tau-microtubule-Dynamin-1 link is specifically at the core of the deleterious effects of elevated Tau levels, it needs to be shown that actin organization is unperturbed as expected, and that the levels of key pre-synaptic proteins (e.g. endophilin) and their sub-synaptic distribution are unaffected by Tau. These issues need to be addressed experimentally.

We did not perform actin imaging for the following reasons: (i) Unlike FTDP mutant tau, wild-type tau cannot bind to actin (Roger et al., 2004, quoted in our manuscript). (ii) The toxic effects of wild-type tau on vesicle endocytosis and synaptic transmission were fully rescued by microtubule-depolymerizing reagent nocodazole (Figure 3). (iii) At the calyx of Held, there is no evidence for vesicle endocytosis to be supported by the F-actin-based endocytic scaffold mechanism involving endophilin, formin or intersectin. On the contrary, F-actin depolymerization with latrunculin treatments (Eguchi et al. 2017 J Neurosci; Piriya Ananda Babu et al. 2020 J Neurosci) or genetic ablation of intersectin (Sakaba et al., 2013 PNAS) has no effect on vesicle endocytosis at the calyx of Held. Endophilin together with synaptojanin reportedly mediates ultra-fast endocytosis and SV uncoating and following acidification, but their deficiency has no effect on clathrin-mediated endocytosis (Watanabe et al., 2018), which is a dominant form of endocytosis at the calyx of Held.

Essential Revisions - Optional Additional Experiments or Text RedactionThe reviewers identified two additional important issues that need to be addressed. If corresponding data are available or can be generated rapidly during the revision phase, they should be included. If this turns out to be problematic, the issues should be covered in the discussion part of the paper - e.g. as an outlook (points 3 and 4) or in the context of defining the general relevance and limitations of the present study (point 4).3. It would be very interesting to assess the effects of elevated Tau levels on other, better characterized microtubule binding proteins in the pre-synapse, e.g. Formins such as mDia, which associate with actin and microtubules and have been implicated in pre-synaptic endocytosis.

Using capacitance measurements, we have tested the effect of the formin inhibitor SMFH2 on vesicle endocytosis at the calyx of Held in slices from post-hearing mice (Figure S5A). Although SMFH2 reportedly inhibits endocytosis at the calyx of Held in slices from pre-hearing rats (P8-P12, Soykan et al, 2017, Neuron), it had no significant effect on exo-endocytosis at the calyx of Held from post-hearing mice (P13-14). Thus, association of formin with MT or F-actin cannot explain the endocytic inhibition by tau (Figure 2 and Figure 3B, explanations in p4, 4^th^ paragraph). These new data are obtained by Satyajit Mahapatra, who is now included as a co-author in our revised manuscript.

Moreover, evidence from neuroendocrine cells indicates that microtubules serve as platforms to direct exo-endocytic coupling by promoting 'hopping' of Clathrin structures, so that the question arises as to whether contributions of such processes to the observed pre-synaptic rundown and endocytic blockade can be ruled out.

We could not reach the paper indicating “hopping” promoted by MTs, but if it is related to SV uncoating, its block can affect vesicle refilling with glutamate, thereby reducing the mean quantal size. However, tau had no such effect with no sign of alteration in the amplitude or frequency of quantal EPSCs (Figure S5B, explained in p4, last paragraph and p5, first paragraph).

4. Given the prominent cortical pathology of Alzheimer's Disease, it would be very informative to assess whether the principal observation that increased levels of soluble Tau sequester Dynamin-1 on microtubules holds true for synapses other than the Calyx of Held, for example central cortical synapses.

This is important but beyond the scope of our present slice model study. We plan to test this in our ongoing tau-propagation culture model.